# Aptamers, a New Therapeutic Opportunity for the Treatment of Multiple Myeloma

**DOI:** 10.3390/cancers14215471

**Published:** 2022-11-07

**Authors:** Ane Amundarain, Fernando Pastor, Felipe Prósper, Xabier Agirre

**Affiliations:** 1Center for Applied Medical Research (CIMA), IDISNA, University of Navarra, 31008 Pamplona, Spain; 2Centro de Investigación Biomédica en Red de Cáncer (CIBERONC), 31008 Pamplona, Spain; 3Hematology Department, Clínica Universidad de Navarra, CCUN, University of Navarra, 31008 Pamplona, Spain

**Keywords:** aptamers, multiple myeloma, targeted therapies

## Abstract

**Simple Summary:**

Multiple Myeloma (MM) remains incurable due to high relapse rates and fast development of drug resistances. Monoclonal antibodies (mAb) have revolutionized MM treatment, opening the door to chemotherapy-free yet curative treatments. Nevertheless, antibody-based therapies face several difficulties which could be overcome by nucleic acid aptamers. Aptamers are short oligonucleotide ligands that bind their targets with great affinity and specificity, and can be easily conjugated to different cargoes for their cell-specific delivery. This review summarizes the aptamers that have been tested in MM so far with promising results, and proposes various strategies for the further development of aptamer-based strategies against MM.

**Abstract:**

Multiple Myeloma (MM) remains an incurable disease due to high relapse rates and fast development of drug resistances. The introduction of monoclonal antibodies (mAb) has caused a paradigm shift in MM treatment, paving the way for targeted approaches with increased efficacy and reduced toxicities. Nevertheless, antibody-based therapies face several difficulties such as high immunogenicity, high production costs and limited conjugation capacity, which we believe could be overcome by the introduction of nucleic acid aptamers. Similar to antibodies, aptamers can bind to their targets with great affinity and specificity. However, their chemical nature reduces their immunogenicity and production costs, while it enables their conjugation to a wide variety of cargoes for their use as delivery agents. In this review, we summarize several aptamers that have been tested against MM specific targets with promising results, establishing the rationale for the further development of aptamer-based strategies against MM. In this direction, we believe that the study of novel plasma cell surface markers, the development of intracellular aptamers and further research on aptamers as building blocks for complex nanomedicines will lead to the generation of next-generation targeted approaches that will undoubtedly contribute to improve the management and life quality of MM patients.

## 1. Multiple Myeloma and Monoclonal Antibodies

Multiple Myeloma (MM) is a clonal B-cell neoplasm characterized by the uncontrolled proliferation and accumulation of malignant plasma cells (PCs) in the bone marrow [1]. Over the past few years, significant progress has been made in understanding MM disease biology and in the development of targeted treatment approaches, leading to deeper and more durable clinical responses and improved MM patient survival [2]. Nevertheless, MM remains an incurable disease due to its high relapse rate and the development of resistant disease [3,4].

Currently, frontline therapies for MM mostly rely on proteasome inhibitors and immunomodulatory drugs (IMiDs) [3]. However, as for many other tumors, the development of monoclonal antibody (mAb) based drugs against specific tumor antigens has led to a paradigm shift in MM treatment, paving the way towards chemotherapy-free yet curative treatments with increased efficacy and better safety profiles [5,6]. In fact, the anti-CD38 mAb Daratumumab has been included as first-line therapy in both transplant eligible and ineligible MM patients, with several other mAb and antibody–drug conjugates (ADC) being approved for refractory patients or being tested in clinical trials (Table 1) [5,6,7]. 

**Table 1 cancers-14-05471-t001:** List of monoclonal antibodies and antibody-drug conjugates which have been approved by the FDA or are in clinical trials for the treatment of MM patients. ADC: antibody–drug conjugate; mAb: monoclonal antibody; ADCC: antibody-dependent cytotoxicity; CDC: complement-dependent cytotoxicity; PCD: programmed cell death; NDMM: newly diagnosed multiple myeloma; RRMM: relapsed/refractory multiple myeloma.

Name	Target	Class	Mechanism	Phase	References
**Daratumumab**	CD38	mAb	ADCC, CDC	Approved alone for RRMM non-responder to proteasome inhibitors and immunomodulatory agents; in combination with lenalidomide and dexamethasone for RRMM or non-autologous stem cell transplant eligible NDMM; in combination with bortezomib, prednisone and melphalan for non-autologous stem cell transplant eligible NDMM, in combination with bortezomib, thalidomide and dexamethasone for autologous stem cell transplant eligible NDMM, in combination with bortezomib/carfilzomib and dexamethasone for RRMM, in combination with pomalidomide and dexamethasone for RRMM with at least one prior treatment with lenalidomide and a proteasome inhibitor	[6,8]
**Isatuximab**	CD38	mAb	ADCC, CDC	Approved in combination with pomalidomide and dexamethasone for RRMM	
**Elotuzumab**	SLAMF7	mAb	ADCC	Approved in combination with lenalidomide, pomalidomide and dexamethasone for RRMM	
**Belantamab mafodotin**	BCMA	ADC (monomethyl auristatin-F)	ADCC, mitosis inhibition by microtubule disruption	Approved in monotherapy for RRMM	
**MEDI2228**	BCMA	ADC (pyrrolobenzodiazepine)	DNA alkylation	Clinical Trial. Phase I	[9]
**Tabalumab**	BAFF	mAb	Soluble and membrane-bound BAFF neutralization	Clinical Trial. Phase II	[10,11]
**Milatuzumab**	CD74	mAb	CD74 antagonist	Clinical Trial. Phase I/II	[12]
**Siltuximab**	IL-6	mAb	IL-6 neutralization	Clinical Trial. Phase II	[13]
**BI-505**	ICAM1	mAb	Macrophage dependent PCD	Clinical Trial. Phase II	[14]
**Indatuximab ravtansine**	CD138	ADC (maytansinoid DM4)	Mitosis inhibition by impediment of tubulin polymerization and microtubule assembly	Clinical Trial. Phase I/IIa	[15]
**Pembrolizumab**	PD-L1	Checkpoint inhibitor		Clinical Trial. Phase III	[16]
**Atezolizumab**	PD-L1	Checkpoint inhibitor		Clinical Trial. Phase Ib	[17]
**Nivolumab**	PD-1	Checkpoint inhibitor		Clinical Trial. Phase I	[18]

Nevertheless, novel approaches developed recently, such as nucleic acid aptamers, also termed “chemical antibodies”, could enable us to expand and overcome the possibilities offered by antibody-based therapies. These aptamers comprise a unique class of biomolecules that are functionally comparable to antibodies but with various advantages for therapeutic application due to their chemical nature, reducing their manufacturing costs and expanding the range of available chemical modifications and conjugations. Furthermore, as their mechanism of action does not depend on immune-mediated cytotoxic effects, aptamers could be still useful in cases where antibody-based therapies have failed due to immune-related resistance mechanisms [8].

## 2. Aptamers

Nucleic Acid Aptamers are short, single-stranded DNA (ssDNA) or RNA molecules that are selected for binding to a specific molecular target. Their defined three-dimensional structures enable the recognition of specific targets with high affinity and specificity [9]. High affinity aptamer generation was initially described in 1990 [10,11], leading to the generation of aptamers against a wide range of targets, including proteins, organic molecules, metal ions, viruses, bacteria, whole cells and even live animals [9]. Currently, the Systematic Evolution of Ligands by Exponential Enrichment (SELEX) described in 1990 continues to be the gold standard in vitro methodology for the generation of both ssDNA and RNA aptamers. SELEX strategy essentially consists of the enrichment of sequences with the ability to bind a given target from an initial random oligonucleotide library (Figure 1). This initial combinatorial library contains a 20–60 nucleotide-long random core flanked by fixed primer regions at 5′ and 3′ [9,12,13], accounting for approximately 10^15^ different molecular species. The tridimensional structure and derived binding properties of each aptamer depend on its nucleotide sequence. Unfortunately, the knowledge for rational in silico design of aptamers towards a given target is still limited [14]; hence, SELEX is useful as a high-throughput screening method, which allows us to simultaneously test billions of different aptamers towards a given target and identify the best binders. For this purpose, the initial aptamer pool is incubated with the purified target protein, bound aptamers are recovered and re-amplified, and then subjected to a new binding round (Figure 1). After various iterative selection rounds, specific aptamers that bind the target will become enriched and will dominate the pool of library sequences, which is then sequenced to determine the nucleotides of the most enriched aptamer families. The final selected aptamers are highly dependent on the selection environment, which is influenced by both experimental conditions and target-inherent properties [9,15,16] and will determine the function and affinity of the selected enriched aptamers. 

Although ssDNA and RNA aptamers are functionally similar, they possess their own advantages and drawbacks. Indeed, the biggest differences reside in their selection protocol during the SELEX process (Figure 1). ssDNA-bound aptamers are amplified through PCR, followed by strand separation to obtain ssDNA aptamers for the next selection round. On the contrary, RNA aptamers must be reverse transcribed for further amplification, followed by in vitro transcription to obtain RNA aptamers for a new selection round. This makes the identification of RNA aptamers more expensive and time consuming than ssDNA aptamers in the SELEX process. RNA aptamers are also less stable than ssDNA aptamers. They offer greater conformational diversity and stronger intra-strand interactions, which can probably increase their binding affinity and specificity [9,17]. Hence, the choice of either strategy will depend on the final requirements of the intended aptamer. Not only the careful choice of oligonucleotide chemistry, but also the inclusion of new methods in aptamer synthesis, technical equipment and analysis, has significantly increased and improved the chance of discovering successful aptamers [9]. Classical SELEX protocols against purified recombinant proteins immobilized in solid-phase matrices (Figure 1) may fail to recognize the native protein conformation expressed on the cell surface. Besides, protein-based SELEX strategy is not applicable for unknown proteins, insoluble proteins or proteins that only function in multi-protein complexes or native conformations. Hence, to overcome this limitation, various “live” SELEX approaches where targets are expressed in their native state have been implemented [9,18]. For example, the whole-cell SELEX approach is based on using living, intact cells to obtain aptamers that specifically bind targets in their native conformation on the cell surface of desired cells, thus enabling the selection even without previous knowledge of the potential target [18]. However, as in vitro selected aptamers may not recapitulate their effects in vivo, live-animal-based SELEX enables the generation of tissue-penetrating aptamers directly within a living animal model of the target disease. The development of SELEX approaches allowing the selection of aptamers in a physiologically appropriate and optimized manner will help to identify aptamers with consistent in vitro and in vivo performance, thus contributing to increased clinical translatability of selected molecules [18].

In comparison to their protein counterpart, aptamers possess many advantageous features for their application as therapeutic agents. Unlike large protein antibodies (150–180 kDa), aptamers have a smaller size (6–30 kDa) and flexible structures, enabling the recognition of inaccessible binding domains for larger antibodies. In addition, smaller aptamers possess increased tissue permeability, which could be especially favorable in solid tumors where antibodies exhibit limited tissue penetration. However, their smaller molecular weight can make aptamers more susceptible to rapid renal clearance and consequent short circulation time in vivo, which could be largely overcome by aptamer chemical modification and conjugation, leading to an improved pharmacokinetic profile and increased half-life in vivo. Aptamers also present various manufacturing benefits, as they can be obtained via chemical solid-phase synthesis, reducing their immunogenicity as well as their production costs and batch-to-batch variation [9,19,20]. As this unique class of biomolecules combines the flexibility of small molecules with the high specificity of antibodies, aptamer-based therapeutics can exploit various approaches: (1) they can serve as agonists to activate the function of target receptors, (2) they can serve as antagonists to block the interaction of disease-associated targets such as receptor–ligand or protein–protein interactions and (3) they can be used as carriers to deliver therapeutic agents to specific cells or tissues. In fact, aptamers are increasingly recognized as one of the best available targeting moieties [21], as their easily modifiable chemical nature enables the conjugation to various therapeutic agents or delivery vehicles for cell-specific targets, leading to increased local concentration and therapeutic efficacy [9]. So far, aptamers have been conjugated to various oligonucleotide therapeutic modalities, such as siRNAs [22], miRNAs [23], anti-miRs [24] or antisense oligonucleotides [25], allowing for successful specific delivery to target cells, which results in gene expression modulation [25] and as a consequence, great applicability for the RNA medicine of the future. Recent studies have also shown the feasibility of generating more complex constructions, such as bivalent and multivalent structures, to achieve different therapeutic effects with the same molecule [21]. Likewise, chemical solid-phase synthesis of aptamers enables their covalent conjugation or physical loading with small organic molecules, polymers, radiopharmaceuticals, or even large enzymes and proteins [9]. Furthermore, beyond direct conjugation of aptamers and therapeutic agents, cell-type specific aptamers are also being exploited to “decorate” organic and inorganic nanocarriers loaded with multiple cargoes [9,26], supporting the development of further aptamer-based nanomedicine.

While multiple proof-of-concept studies have demonstrated the potential of aptamers and derivatives as therapeutic agents, nowadays, the development of clinically effective aptamers still lags far behind that of therapeutic antibodies [9,27] for various reasons. As aforementioned, their poor pharmacokinetic profile could be improved with chemical modifications and conjugation; however, the toxicological information regarding aptamers and subsequent modifications in humans is still very limited [9]. Moreover, in contrast with well-known antibodies, aptamers still lack the necessary investment, education and commercial infrastructure for their clinical translation and widespread distribution. Nevertheless, given that patients with MM urgently need new therapeutic approaches and that the problems faced by antibody-based drugs could be overcome by aptamers, we believe that this technology can greatly contribute to the development of novel targeted therapies against MM. We expect that further research into the aptamer field will lead to the development of next generation aptamer-based approaches for an improved management and treatment of MM.

## 3. Current Aptamers for MM Precision Medicine

As for antibodies, an ideal target for MM-specific aptamer development would be one with a high and uniform expression in MM cells to have sufficient efficacy, while having a negligible to low expression on other normal cells to avoid on-target off-tumor effects. Hence, most aptamers being tested for MM are thus aimed at the same well-known molecules for which antibodies have been developed, showing some promising preclinical results for several of these aptamers (Figure 2; Table 2):

**Aptamer against AXII**: Annexin A2 (AXII) is a calcium-dependent, phospholipid binding member of the annexin family. AXII is overexpressed in MM plasma cell membranes, with its expression being negatively correlated with patient survival [28]. The interaction of AXII with its receptor AXIIR enhances MM cell adhesion and growth in the BM microenvironment, potentially supporting the homing and growth of MM cells in the BM. AXII can be secreted by various cell types in the BM, promoting MM cell growth by creating a pro-tumorigenic niche, thus, targeting the AXII/AXIIR axis represents an attractive approach for the development of therapies targeted at the MM niche [29]. Zhou et al. [30] identified a ssDNA aptamer (wh6) that was able to bind AXII in the low nanomolar range through nine rounds of protein-based SELEX. They showed that the aptamer was able to specifically bind MM cells expressing AXII both in vitro and in vivo, and it could inhibit the AXII induced adhesion and progression in MM cell lines, indicating the suitability of the wh6 aptamer for targeted MM treatment.**BCMA targeted aptamer:** B-cell maturation antigen (BCMA) is a member of the tumor necrosis factor (TNF) receptor superfamily, which is preferentially expressed by late-stage B lymphocytes while showing minimal expression in hematopoietic stem cells [31]. Under physiological conditions, the binding of its specific ligands BAFF and APRIL induces the activation of both canonical and non-canonical NF-κB pathways, promoting long-lived plasma cells survival. However, BCMA overexpression and increased activation are associated with MM progression in terms of the upregulation of NF-κB pathways and subsequent overexpression of critical genes for MM growth and survival [31,32]. In this direction, Catuogno et al. [32] selected a BCMA-targeted internalizing RNA aptamer (apt69.T) through a variation of the cell-SELEX approach. In vitro approaches using MM cell lines showed that the selected apt69.T aptamer was able to readily bind BCMA and inhibit the APRIL dependent downstream signaling pathway. Furthermore, it was able to internalize rapidly and successfully deliver therapeutic oligonucleotides to MM cells. For that, the BCMA aptamer was conjugated to miRNA and miRNA antagonists using a sticky-end based approach, leading to consequent upregulation of miR-137 and downregulation of miR-222 in MM cell lines. The upregulation of the tumor suppressor miR-137 was able to reduce MM cell viability, highlighting the feasibility of using MM-specific aptamers for the effective delivery of therapeutic oligonucleotides to MM cells.**Aptamer against C-MET:** C-MET is a transmembrane tyrosine kinase known to be the receptor of the hepatocyte growth factor (HGF) cytokine. In MM, C-MET expression gradually increases during disease development, its high expression being correlated with poor MM patient outcomes [33]. Upon HGF binding, C-MET dimerizes, resulting in kinase auto-phosphorylation and the creation of a multi-substrate docking site necessary for the induction of downstream signaling cascades which ultimately contribute to MM development by promoting cell growth, migration and angiogenesis while inhibiting apoptosis [33]. SL1 [34] is the truncated version of the original CLN0003 ssDNA aptamer, which was selected against purified C-MET through a filter SELEX approach for the recognition of C-MET overexpressing tumors [35]. Accordingly, Zhang et al. [33] demonstrated that targeting C-MET via the SL1 aptamer [34,35] could be a potential therapeutic approach in MM, showing that SL1 was able to inhibit HGF-dependent C-MET signaling and suppress MM cell growth in vitro. Furthermore, SL1 showed synergism with bortezomib, highlighting the potential for novel combination therapies in MM.**Conjugated CD38-doxorubicin aptamer:** CD38 is a cell surface glycoprotein which is highly and homogeneously expressed in MM cells, with minimal expression on normal myeloid and lymphoid cells. This highly versatile molecule contributes to MM development by acting as a receptor for proliferative signaling, as an adhesion molecule or as an ectoenzyme in the catabolism of NAD+ and NADP [36,37]; thus, in recent years, it has become one of the main targets for anti-MM targeted therapy development. Wen et al. [38] were able to identify a CD38 specific ssDNA aptamer via a hybrid protein- and cell-based SELEX approach. This aptamer was subsequently non-covalently conjugated to doxorubicin for the generation of CD38-specific aptamer–drug conjugates (ApDC). The ApDCs were readily internalized by MM cells, and after a pH-dependent release of the cargo in lysosomes, doxorubicin was able to exert its specific antitumor activity by inhibiting tumor growth without toxicity in both MM in vitro and in vivo models.**RNA aptamer for CXCL12:** CXCL12, also known as stromal cell-derived factor-1 (SDF-1), is a chemoattractant chemokine that, upon binding to its receptors CXCR4 and CXCR7, induces the adhesion and homing of MM cells to the protective BM niche [39,40], and therefore, it is considered one of the major players in cell adhesion-mediated drug resistance (CAM-DR) [40]. MM cells present high levels of CXCL12, CXCR4 and CXCR7; therefore, CXCL12 neutralization represents an attractive option to modulate the BM niche for MM therapy and overcome CAM-DR [39,40].

NOX-A12 (olaptesed pegol) is the RNA spiegelmer aptamer that binds and antagonizes CXCL12, which was identified by protein SELEX against the D enantiomer of the natural L-CXCL12 protein [39]. Spiegelmers are synthetic RNA aptamers in which the natural D-configuration ribonucleotides have been replaced by their enantiomer (mirror-image) L-ribonucleotides. Like aptamers, they bind their targets with high affinity and specificity; however, due to the presence of unnatural ribonucleotides, they are not susceptible to nuclease degradation or hybridization to native oligonucleotides, and they do not exert immune responses [41,42]. In a study from 2014, Roccaro el al. [39] showed that in vivo CXCL12 neutralization by NOX-A12 could reduce MM cell homing and growth, thereby inhibiting disease progression by disrupting BM colonization by MM cells and inducing the release of MM plasma cells to circulation. Likewise, a phase IIa clinical trial from 2014 combining NOX-A12 with bortezomib and dexamethasone (NCT01521533) in MM refractory patients showed that NOX-A12 was able to mobilize plasma cells without toxic effects [43]. Nevertheless, the absence of more recent studies involving NOX-A12 in MM suggest that further research on this molecule in MM has been discontinued, despite the acceptable results obtained in the clinical trial. However, there are currently two open clinical trials with NOX-A12 in solid tumors: the GLORIA (NCT04121455) phase I/II trial in glioblastoma and the OPTIMUS (NCT04901741) phase II trial in metastatic pancreatic cancer, suggesting that NOX-A12 in combination with other therapies may still contribute to the improved management of human cancer patients.

**Table 2 cancers-14-05471-t002:** Current aptamers for MM precision medicine.

Target	Class	Identification	Internalization in mm Cells	Therapeutic Application	Effect	Stage	Reference
**AXII**	ssDNA	Recombinant protein SELEX	Not determined	Aptamer alone.	Inhibition of MM cell-line adhesion and proliferation	Preclinical	[40]
**BCMA**	RNA	Cell-SELEX	Yes	Aptamer alone.	Inhibition of BCMA pathway in vitro.	Preclinical	[42]
				Aptamer miRNA (miR-137) chimera.	Upregulation of tumor suppressor miR-137 leading to reduced viability in vitro.		
				Aptamer anti-miRNA (anti-miR-222) chimera.	Inhibition of oncogenic miR-222 in vitro.		
**C-MET**	ssDNA	Recombinant protein SELEX	Not determined	Aptamer alone.	Suppression of HGF-induced C-MET activation, inhibition of MM cell line proliferation and increased apoptosis, inhibition of cell migration and adhesion.	Preclinical	[20,43,44]
				Combination therapy with bortezomib.	Synergy with bortezomib.		
**CD38**	ssDNA	Hybrid protein and cell SELEX	Yes	Aptamer-Doxorubicin conjugate.	Inhibition of MM cell-line proliferation in vitro, tumor inhibition in xenograft models	Preclinical	[45]
**CXCL12 (NOX-A12)**	spiegelmer	Protein SELEX against target enantiomer	No	Aptamer alone.	Inhibition of CXCR4 and CXCR7 activity in vitro, reduction of tumor growth in vivo, release of plasma cells into circulation in vivo.	Preclinical	[46,47,48]
				Combination therapy with bortezomib.	Synergy with bortezomib in vivo.	Preclinical	
				Combination therapy with dexamethasone and bortezomib.	Plasma cell mobilization.	Clinical Trial. Phase II completed (NTC01521533)	

## 4. Future Directions

Although few aptamers have been tested in MM so far, we believe that these promising preclinical and clinical results merit further exploration of the aptamers above. Likewise, if the target is relevant for MM disease, we consider that exploring the performance of known aptamers with promising results in other diseases [9,20,36] could also be worth the effort.

In comparison to other hematological diseases in which aptamers have been further explored [20,36], aptamer research in MM has probably only touched the tip of the iceberg. The promising results reviewed above encourage further research in the field. In this section, we will review three novel possibilities with a great potential to expand MM aptamer development, hoping they could boost MM aptamer research and lead to a novel generation of targeted therapies for MM treatment.

**Novel surface targets:** Extensive development in sequencing approaches has enabled the unbiased study of genomes, epigenomes, transcriptomes and epitranscriptomes of healthy and diseased states, offering invaluable insight into altered genes and pathways and enabling the development of novel targeted drugs. However, the same depth of knowledge has not yet been achieved for the tumor cell surface, whose composition remains poorly explored [44]. The cell surface proteotype (surfaceome) encompasses all plasma membrane proteins with at least one extracellularly exposed amino acid, which can be decorated with various post-translational modifications (PTMs) and can interact with proteins or ligands in the same cell (cis), in other cells (trans), with extracellular matrix components, drugs and hormones [49]. All these interactions shape the relationship between the cancer cell and the local microenvironment. Recent proteomic studies have shown the degree of surfaceome remodeling in cancer and tumor resistance to front-line chemotherapeutic agents [45,46,50], leading to the discovery of novel disease and resistance biomarkers which will undoubtedly broaden the range of therapeutic targets.

Ferguson et al. [46] have recently performed the first comprehensive MM plasma cell surfaceome study at baseline, in drug resistance and in response to acute drug treatment. Through an unbiased proteomic approach, they were able to identify novel yet unexplored MM potential therapeutic targets such as CCR10, TXNDC11 and LILRB4. Furthermore, they described the surfaceome remodelation occurring upon short- and long-term treatments with bortezomib and lenalidomide, providing novel targets for combination therapeutic approaches to overcome these resistances. Undoubtedly, the deep knowledge of tumor surfaceome will contribute to the rational design of a new generation of aptamers against highly specific, yet unknown targets. Through further method refinement, the possibility of extending this proteomic profiling to individual disease samples [46] would enable the identification of patient-specific biomarkers, which could be readily targeted by patient-specific aptamer-based approaches.

**Intracellular aptamers:** So far, aptamer development has mainly focused on cell-surface proteins. However, theoretically, aptamers can be selected against virtually many proteins with great specificity, and are even able to distinguish between closely related molecules such as conformational isomers, targets with different functional groups or proteins with point mutations [9]. Furthermore, as RNA aptamers can be genetically encoded as transgenes for endogenous cellular expression, it is not surprising that the development of intracellular aptamers (intramers) has started to gain relevance only recently [47,48].

Transcription factors (TF) have pivotal roles in regulating gene expression both in health and disease, and their deregulation has been described as a key event in cancer development [51]. Thus, the inhibition of deregulated transcription-factor-dependent transcriptional programs has emerged as an attractive therapeutic approach in the last few years, even for MM treatment [52]. For example, the addition of anti forkhead box M1 (FOXM1) [53] and heat shock factor (HSF1) [54] aptamers to breast and cervical cancer cell lines showed the inhibition of their downstream transcriptional programs and led to impaired cell growth and apoptosis, respectively. Likewise, other groups were able to generate aptamers that could act as molecular decoys by preventing a given transcription factor to bind DNA. For example, Maher III et al. identified RNA aptamers that were able to competitively inhibit DNA binding by NF-κB, thereby acting as molecular decoys to inhibit NF-κB-dependent transcriptional activation [55,56,57]. Likewise, Barton et al. [58] identified RNA aptamers that were able to disrupt the formation of DNA-RUNX1-CBFβ complex competitively.

Along with deregulated transcription factors, disease-specific isotypes and point mutations also represent attractive targets for targeted anti-cancer effects. It is well described how point mutations in tumor-suppressor proteins cause cancer [59]. In fact, point mutations in the key tumor suppressor p53 can be found in more than half of human cancers [60]. p53R175 constitutes a mutational hotspot, where the amino acid mutation p53R175H can lead to p53 WT function inhibition in a dominant negative manner with the consequent anti-apoptotic and pro-metastatic effects in various cancers [60]. Chen et al. [60] applied the contrast-SELEX strategy to identify an RNA aptamer capable of specifically recognizing the p53 protein harboring R175H mutation. The aptamer was able to partially rescue p53 WT function, leading to reduced growth and migration and increased apoptosis in lung cancer cell lines and xenograft models.

Another hallmark of cancer constitutes the translational deregulation of pro-oncogenic mRNAs, where such mRNAs depend on the eukaryotic initiation factor 4A (eIF4A) for efficient translation [47]. In fact, the eIF4A inhibition could synergize with dexamethasone in MM, leading to reduced translation of master MM transcription factors such as MYC [61]. Oguro et al. [62] developed an RNA aptamer that was able to block cap-dependent translation by blocking eIF4A ATP hydrolysis. Intriguingly, they postulated that their aptamer was able to inhibit the conformational change induced upon ATP binding to eIF4A, which is responsible for its helicase activity and subsequent aberrant translation in cancer.

Globally, these results show the feasibility of modulating intracellular protein activity via intramers. Although barely explored yet, intramers can contribute to the expansion of druggable targets by allowing the inhibition of deregulated transcription factors or disease-specific isotypes and mutated proteins which could have not been targeted by other pharmacological approaches for a long time [48,63].

**Novel aptamer-based therapeutic agents:** The fact that aptamers can be easily conjugated to a wide range of therapeutic approaches highlights the enormous potential of aptamers as building blocks for the generation of complex next-generation nanomedicines. Regarding MM, only the CD38-doxo and BCMA-antimiR/miRNA aptamer–drug conjugates have been tested so far in preclinical stages. However, we expect that recent advances in aptamer-based multivalent structures and nanomedicines will expand to MM research [9,21,26], enabling the development of new-generation drugs with increased therapeutic effect and reduced toxicities.

## 5. Concluding Remarks

Aptamers comprise a unique class of biomolecules that combine the flexibility of small molecules with the high specificity of antibodies. In MM, aptamer-based approaches are still taking their first steps, but initial encouraging results have revealed their enormous potential to target the disease from multiple different angles. Still, their successful clinical translatability requires deeper knowledge of their pharmacological performance and the development of a whole commercial infrastructure. Nevertheless, we expect that further research in the field will lead to the development of next-generation aptamers with increased efficacies and reduced toxicities, which will undoubtedly contribute to improving MM patient survival and life quality.

## Figures and Tables

**Figure 1 cancers-14-05471-f001:**
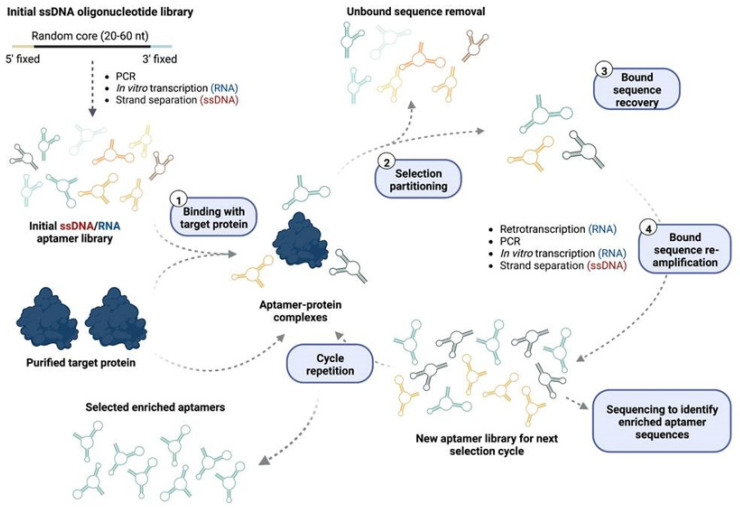
Overview of purified protein-based SELEX. The initial RNA/ssDNA aptamer pool is generated from a commercial ssDNA oligonucleotide library consisting of a central random core sequence flanked by fixed 5′ and 3′ primer binding regions to allow library amplification. To obtain RNA/ssDNA aptamers for the first selection cycle, the library is amplified by PCR and subjected to in vitro transcription or strand separation, respectively. A purified protein-based SELEX approach has four key steps: (**1**) the aptamer library is initially incubated with the purified target protein and (**2**) bound sequences are isolated from the unbound ones via different partitioning strategies. The bound sequences are then (**3**) recovered and (**4**) re-amplified to obtain the new aptamer library for the next selection cycle. For RNA aptamers, recovered RNA aptamers must be retrotranscribed for subsequent PCR amplification, and the amplification product is then converted into RNA again via in vitro transcription. For ssDNA, recovered aptamers are directly amplified by PCR and ssDNA aptamers are obtained by strand separation. After n selection cycles, the aptamer pool is sequenced to identify the enriched aptamer sequences. Image made in ©BioRender.

**Figure 2 cancers-14-05471-f002:**
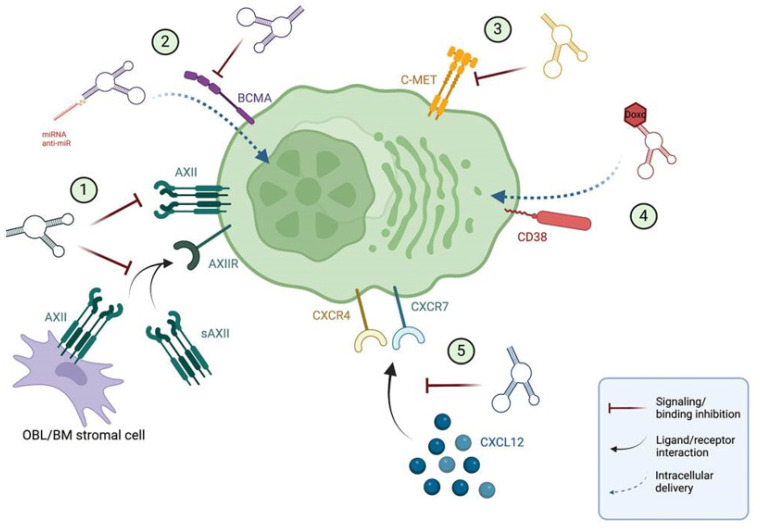
Current aptamers for precision medicine in MM. (**1**) Aptamer against AXII, (**2**) BCMA targeted aptamer, (**3**) aptamer against C-MET, (**4**) conjugated CD38-doxorubicin aptamer and (**5**) NOX-A12 RNA aptamer for CXCL12. All the aptamers except the NOX-A12 spiegelmer are directed towards receptors expressed in the MM plasma cell membrane. Receptor–ligand interactions are depicted with black arrows, while the mechanism of action of aptamers is depicted with red arrows for antagonist aptamers, and blue arrows for cargo delivery aptamers. BM: bone marrow; Doxo: doxorubicin; OBL: osteoblast; sAXII: soluble annexin A2. Image made in ©BioRender.

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
