# Peer review of "Aptamers, a New Therapeutic Opportunity for the Treatment of Multiple Myeloma"

_cancers, 2022, doi:10.3390/cancers14215471_

Round 1
Reviewer 1 Report
The authors presented a fully comprehensive review of aptamers as a new therapeutic option in multiple myeloma. The structure and biology of aptamers is described in details clearly and as concise as possible.
Furthermore, the clinical implementation of this knowledge especially in the sight of the novel therapeutic possibilities was addressed as well.
Different aptamers for MM precision medicine are have been discussed in details and accompanied by excellent schematic presentation.
However, some aptamers are directed against well known targets like CD38 or BCMA. Since there are already approved therapies using same targets (like Daratumumab, Tafasitamab, Belantaman or even CARTs), it would be interesting to know, weather aptamers could show activity after failure of above mentioned therapeutics.
In my opinion, the review is well presented and clearly written including relevant data on pathogenesis of myeloma development. Minor revision is recommended.
Author Response
We would like to thank the reviewer for the interest shown in our review and the positive feedback on the relevance of the topic discussed in the review.
Regarding his/her question, we agree that it would be very interesting to know if aptamers could still maintain their activity when patients become refractory to other therapeutic approaches against the same target receptors. Therefore, we have tried to address that question by conducting a bibliographic search on the latest data on the mechanisms of resistance to different immunotherapies developed against the same target molecules.
Beyond immunotherapy, aptamers represent a novel therapeutic tool to expand the therapeutic strategies for MM, because unlike monoclonal antibodies and CAR-T cells, aptamers do not depend on the immune system to trigger an antitumor response. In fact, the greatest advantage of aptamers resides in their chemical nature, which allows their conjugation to a wide variety of effector molecules beyond their possible therapeutic effect by themselves. Therefore, aptamers could be used as therapeutic molecules as well as specific carrier molecules of other therapeutic drugs, and hence, the anti-tumoral of the aptamer will depend largely on the cargo molecules it carries.
Therapy resistance is a complex mechanism that involves both the nature of the target and the therapeutic strategy used. Indeed, several mechanisms underlying CD38 and BCMA-directed therapy resistance have been described. One of these mechanisms includes target downregulation of CD38 and BCMA in response to daratumumab [1] monoclonal antibody or BCMA CAR-T [2] directed therapies. In these cases, the aptamers would not be useful after the failure of the aforementioned therapies since the target would be down-regulated and, therefore, no response should be expected. Nevertheless, this effect is considered rare in the case of BCMA as the receptor is necessary for plasma cell survival [3], and thus, the viability of sequential anti-BCMA agents remains unclear [4]. Equally, it has been postulated that in the case of CD38 [1], combination with drugs that can re-establish target expression, such as ATRA or Panobinostat, could help to overcome resistances mediated by target downregulation. Furthermore, it is still unknown if aptamer-based therapies without previous targeting of the same receptor will trigger target downregulation too.
On the other hand, as aptamers are not antibodies, they should not be affected by resistances developed against Fc dependent immune effector mechanisms [1]. Equally, they should not be affected either by CAR-T cell specific resistance mechanisms such as T-cell exhaustion or the generation of anti scFv antibodies [3], hence, theoretically, aptamers against these same targets could still be effective after prior treatment failure. And regarding recently approved ADCs such as belantamab mafodotin, no resistance mechanisms have been described in MM so far, hence, the suitability of sequential anti-BCMA therapies after belantamab mafodotin remains unknown.
Unfortunately, aptamers in MM are still at a very experimental initial phase, and neither of the aptamers against the mentioned targets (BCMA and CD38) have reached clinical trials yet. Therefore, it is currently impossible to evaluate the activity of aptamers in such situations, and the ideas discussed above are all hypothetical situations which require functional in vitro and in vivo validations and studies related to their mechanism of action. However, we believe that further research with aptamers will help to address all these questions.
In response to the issue raised by the reviewer, we have found it relevant to include a reference in the revised version of the manuscript to the different modes of action of aptamers and antibodies, highlighting that aptamers could still be useful in some cases after antibody-based therapy failure:
Manuscript, “Multiple Myeloma and monoclonal antibodies” section, page 3:
“Nevertheless, novel approaches developed recently, like nucleic acid aptamers, also termed as “chemical antibodies”, could enable to expand and overcome the possibilities offered by antibody-based therapies. These aptamers comprise a unique class of biomolecules functionally comparable to antibodies but with various advantages for therapeutic application due to their chemical nature, reducing their manufacturing costs and expanding the range of available chemical modifications and conjugations. Furthermore, as their mechanism of action does not depend on immune-mediated cytotoxic effects, aptamers could be still useful in cases where antibody-based therapies have failed due to immune-related resistance mechanisms[8].”
Reviewer 2 Report
In the present review, the authors summarize the several aptamers that have been tested against MM specific targets with promising results, establishing the rationale for the further development of aptamer-based strategies against MM. The topic is scientifically relevant and innovative in Multiple Myeloma therapeutic armamentarium. I think that it can add a great contribution to this field, since several proof-of-concept studies have demonstrated the potential of aptamers but ongoing efforts are crucial to estabilish their role as a valid molecule such as antobodies. Therefore, I think that this review resume adequately the state-of-the-art of the literature regarding the role of aptamers in multiple myeloma.
Author Response
We thank the reviewer for his/her interest in our manuscript and his/her valuable remarks.
Reviewer 3 Report
General comment:
Authors discussed that nucleic acid aptamers are short, single-stranded DNA (ssDNA) or RNA molecules that are selected for binding to a specific molecular target. Most aptamers being tested for MM are aimed at the same well -known molecules for which antibodies have been developed. In MM, aptamer-based approaches are still taking their first steps, but initial encouraging results have revealed 358 their enormous potential to target the disease from multiple different angles.
This report is considered as important information for physicians. If the authors could add information as below, this paper would be more informative.
Specific comment:
1. Since aptamers has no ADCC or CDC activity, isn't it less effective than the antibody?
2. T-cell–redirecting bispecific antibodies have been reported to be more effective than ADCs in the treatment of multiple myeloma. Is aptamer-drug conjugates less effective than bispecifuc antibodies?
3. The paper on NOX-A12 was reported in 2014. Any new reports about NOX-A12?
Author Response
We would like to thank the reviewer for the thoughtful comments raised above and we understand reviewer’s concern about effectivity of aptamers in comparison to antibodies. Hence, in an effort to clarify the specific suggested points, we have included a detailed answer to the raised questions.
Specific comment:
- Since aptamers has no ADCC or CDC activity, isn't it less effective than the antibody?
- T-cell–redirecting bispecific antibodies have been reported to be more effective than ADCs in the treatment of multiple myeloma. Is aptamer-drug conjugates less effective than bispecifuc antibodies?
Regarding questions 1 and 2, as aptamers are conceptually similar to antibodies in the sense that they also enable target specific recognition, we understand the relevance of comparing them to the results obtained so far with monoclonal antibodies and derived molecules. In fact, a recent excellent review [5] has addressed the role of aptamer-based approaches in comparison to antibody-based therapies for cancer immunomodulation.
Furthermore, we would like to highlight that aptamers comprise a different class of therapeutic molecules, and therefore, their effectivity cannot be directly compared to that of antibodies. We believe that the main advantage of aptamers consists in their chemical nature, implying that aptamers are non-immunogenic molecules with reduced production costs and enhanced conjugation options [5]. In fact, aptamers enable the construction of complex multi-effector molecules [6–9], which undoubtedly require further research for their translation to the clinic, but offer the possibility to the construction of next-generation engineered biomolecules to improve the treatment and response of patients with MM or other types of human tumors. Nevertheless, all these molecules are still at preclinical phases and therefore, their therapeutic potencies in patients are still unknown.
Concerning question 1, as the reviewer indicates, aptamers are not able to trigger immune effector cytotoxic mechanisms [1] such as ADCC or CDC. However, we believe that this does not necessarily mean that aptamers are less effective, mainly because aptamers exert their antitumor effect through other mechanisms. Several studies in other diseases [10] and tumors [11] beyond MM have shown the effectivity of aptamers alone as effector molecules by themselves [5], or as delivery molecules of complex therapeutic drugs [7]. Therefore, as aptamers are novel tools that enable the specific delivery of other novel targeted approaches, we believe that the measurement of aptamer effectivity will mainly depend on the attached cargo.
For question 2, it is undeniable that bispecific and trispecific antibodies represent the next generation of immunotherapies with increased efficacies by enabling the engagement of the tumoral plasma cell and immune effector cells leading to tumor destruction [12]. Nevertheless, systemic administration of these therapies faces dose-limiting side-effects [5], which could be overcome by alternative approaches such as immunomodulatory bispecific Aptamers (bsApt) reviewed by Thomas BJ, et al. [5]. Still, the majorities of these bsApt have been developed as proof-of-concept approaches and despite good performance in pre-clinical set-ups, still require further efforts for their translation into the clinic and the evaluation of their effectivity in patients.
In summary, the clinical development of aptamers still lags far behind that of antibodies, so significantly more research will be needed to address the interesting questions raised by the reviewer. Nevertheless, we believe that aptamers can provide a complementary approach for the development of next-generation targeted combination drug regimens with increased efficacy and improved safety profile.
- The paper on NOX-A12 was reported in 2014. Any new reports about NOX-A12?
We would like to thank the reviewer for his interest in the NOX-A12 molecule and we agree that it is an interesting molecule as it is the first aptamer in clinical trials for the treatment of MM.
As mentioned by the reviewer, the results of the phase IIa clinical trial (NCT01521533) combining NOX-A12 with Bortezomib and Dexamethasone in patients with refractory MM was reported in 2014 [13]. Hence, we have carried out a detailed and updated bibliographical investigation of the NOX-12 molecule to elucidate its current status and its applications. In this literature search, we have not found current preclinical papers or clinical trials involving NOX-A12 in MM, suggesting that further research on NOX-A12 in this neoplasm may have been discontinued despite acceptable results in the phase IIa clinical trial. However, there are currently two open clinical trials with NOX-A12 in solid tumors, suggesting that NOX-A12 in combination with other therapies may still contribute to the improved management of human cancer patients:
1) GLORIA (NCT04121455) phase I/II trial with NOX-A12 in combination with irradiation and bevacizumab/pembrolizumab for glioblastoma patients with unmethylated MGMT.
2) OPTIMUS (NCT04901741) phase II trial with NOX-A12 in combination with Pembrolizumab and Nanoliposomal Irinotecan/5-FU/Leucovorin or Gemcitabine/Nab-paclitaxel in Microsatellite-stable Metastatic Pancreatic Cancer Patients.
Nevertheless, it should be noted that NOX-A12 is a neutralizing aptamer for the soluble CXCL12 chemokine, with a comparable role to other CXCL12 inhibitors. Therefore, NOX-A12 does not offer the cell-specific targeting and delivery opportunities offered by other aptamers which are able to bind membrane receptors and internalize into target cells. This implies that the application of this NOX-A12 aptamer is more limited in comparison with other aptamers that can be used as tools for delivery to specific target cells and for functionalization of various novel therapeutic constructs.
Hence, following reviewer’s advice, we have included this information in the revised version of the manuscript to give a more complete and actualized information regarding NOX-A12 in MM:
Manuscript, “Current aptamers for MM precision medicine - RNA aptamer for NOX-A12” section, pages 8-9:
“NOX-A12 (olaptesed pegol) is the RNA spiegelmer aptamer that binds and antagonizes CXCL12, which was identified by protein SELEX against the D enantiomer of the natural L-CXCL12 protein[39]. Spiegelmers are synthetic RNA aptamers where the natural D-configuration ribonucleotides have been replaced by their enantiomer (mirror-image) L-ribonucleotides. Like aptamers, they bind their targets with high affinity and specificity, however, due to the presence of unnatural ribonucleotides, they are not susceptible to nuclease degradation or hybridization to native oligonucleotides and they do not exert immune responses[41,42]. In a study from 2014, Roccaro el al.[39] showed that in vivo CXCL12 neutralization by NOX-A12 could reduce MM cell homing and growth, thereby inhibiting disease progression by disrupting BM colonization by MM cells and inducing the release of MM plasma cells to circulation. Likewise, a phase IIa clinical trial from 2014 combining NOX-A12 with bortezomib and dexamethasone (NCT01521533) in MM refractory patients showed that NOX-A12 was able to mobilize plasma cells without toxic effects[43]. Nevertheless, the absence of more recent studies involving NOX-A12 in MM suggest that further research on this molecule in MM has been discontinued despite the acceptable results obtained in the clinical trial. However, there are currently two open clinical trials with NOX-A12 in solid tumors: the GLORIA (NCT04121455) phase I/II trial in glioblastoma and the OPTIMUS (NCT04901741) phase II trial in metastatic pancreatic cancer, suggesting that NOX-A12 in combination with other therapies may still contribute to the improved management of human cancer patients.”
REFERENCES
- Franssen, L.E.; Stege, C.A.M.; Zweegman, S.; van de Donk, N.W.C.J.; Nijhof, I.S. Resistance Mechanisms towards CD38−directed Antibody Therapy in Multiple Myeloma. J. Clin. Med. 2020, 9, doi:10.3390/jcm9041195.
- Samur, M.K.; Fulciniti, M.; Aktas Samur, A.; Bazarbachi, A.H.; Tai, Y.T.; Prabhala, R.; Alonso, A.; Sperling, A.S.; Campbell, T.; Petrocca, F.; et al. Biallelic Loss of BCMA as a Resistance Mechanism to CAR T Cell Therapy in a Patient with Multiple Myeloma. Nat. Commun. 2021, 12, 1–7, doi:10.1038/s41467-021-21177-5.
- Manier, S.; Ingegnere, T.; Escure, G.; Prodhomme, C.; Nudel, M.; Mitra, S.; Facon, T. Current State and Next-Generation CAR-T Cells in Multiple Myeloma. Blood Rev. 2022, 54, doi:10.1016/j.blre.2022.100929.
- Vaxman, I.; Abeykoon, J.; Dispenzieri, A.; Kumar, S.K.; Buadi, F.; Lacy, M.Q.; Dingli, D.; Hwa, Y.; Fonder, A.; Hobbs, M.; et al. “Real-Life” Data of the Efficacy and Safety of Belantamab Mafodotin in Relapsed Multiple Myeloma—the Mayo Clinic Experience. Blood Cancer J. 2021, 11, 196, doi:10.1038/s41408-021-00592-3.
- Thomas, B.J.; Porciani, D.; Burke, D.H. Cancer Immunomodulation Using Bispecific Aptamers. Mol. Ther. - Nucleic Acids 2022, 27, 894–915, doi:10.1016/j.omtn.2022.01.008.
- Porciani, D.; Cardwell, L.N.; Tawiah, K.D.; Alam, K.K.; Lange, M.J.; Daniels, M.A.; Burke, D.H. Modular Cell-Internalizing Aptamer Nanostructure Enables Targeted Delivery of Large Functional RNAs in Cancer Cell Lines. Nat. Commun. 2018, 9, doi:10.1038/s41467-018-04691-x.
- Li, S.; Jiang, Q.; Liu, S.; Zhang, Y.; Tian, Y.; Song, C.; Wang, J.; Zou, Y.; Anderson, G.J.; Han, J.Y.; et al. A DNA Nanorobot Functions as a Cancer Therapeutic in Response to a Molecular Trigger in Vivo. Nat. Biotechnol. 2018, 36, 258–264, doi:10.1038/nbt.4071.
- Nuzzo, S.; Roscigno, G.; Affinito, A.; Ingenito, F.; Quintavalle, C.; Condorelli, G. Potential and Challenges of Aptamers as Specific Carriers of Therapeutic Oligonucleotides for Precision Medicine in Cancer. Cancers (Basel). 2019, 11.
- Esposito, C.L.; Catuogno, S.; Condorelli, G.; Ungaro, P.; De Franciscis, V. Aptamer Chimeras for Therapeutic Delivery: The Challenging Perspectives. Genes (Basel). 2018, 9, doi:10.3390/genes9110529.
- Zhou, J.; Rossi, J. Aptamers as Targeted Therapeutics: Current Potential and Challenges. Nat. Rev. Drug Discov. 2017, 16, 181–202, doi:10.1038/nrd.2016.199.
- Shigdar, S.; Schrand, B.; Giangrande, P.H.; de Franciscis, V. Aptamers: Cutting Edge of Cancer Therapies. Mol. Ther. 2021, 29, 2396–2411, doi:https://doi.org/10.1016/j.ymthe.2021.06.010.
- Lancman, G.; Sastow, D.L.; Cho, H.J.; Jagannath, S.; Madduri, D.; Parekh, S.S.; Richard, S.; Richter, J.; Sanchez, L.; Chari, A. Bispecific Antibodies in Multiple Myeloma: Present and Future. Blood Cancer Discov. 2021, 2, 423–433, doi:10.1158/2643-3230.BCD-21-0028.
- Ludwig, H.; Weisel, K.; Petrucci, M.T.; Leleu, X.; Cafro, A.M.; Laurent, G.; Zojer, N.; Foa, R.; Greil, R.; Yakoub-Agha, I.; et al. Final Results from the Phase IIa Study of the Anti-CXCL12 Spiegelmer® Olaptesed Pegol (NOX-A12) in Combination with Bortezomib and Dexamethasone in Patients with Multiple Myeloma. Blood 2014, 124, 2111, doi:10.1182/blood.V124.21.2111.2111.
Round 2
Reviewer 3 Report
I am pleased to tell you that the revised manuscript has been satisfactory improved.
I am happy to accept without revision.